# Physiological constraint on acrobatic courtship behavior underlies rapid sympatric speciation in bearded manakins

Meredith C Miles[1], Franz Goller[2,3], Matthew J Fuxjager[1]*

[1]Wake Forest University, Winston-Salem, United States; [2]University of Utah, Salt Lake City, United States; [3]Institute for Zoophysiology, University of Münster, Germany

**Abstract** Physiology's role in speciation is poorly understood. Motor systems, for example, are widely thought to shape this process because they can potentiate or constrain the evolution of key traits that help mediate speciation. Previously, we found that Neotropical manakin birds have evolved one of the fastest limb muscles on record to support innovations in acrobatic courtship display (Fuxjager et al., 2016a). Here, we show how this modification played an instrumental role in the sympatric speciation of a manakin genus, illustrating that muscle specializations fostered divergence in courtship display speed, which may generate assortative mating. However, innovations in contraction-relaxation cycling kinetics that underlie rapid muscle performance are also punctuated by a severe speed-endurance trade-off, blocking further exaggeration of display speed. Sexual selection therefore potentiated phenotypic displacement in a trait critical to mate choice, all during an extraordinarily fast species radiation—and in doing so, pushed muscle performance to a new boundary altogether.
DOI: https://doi.org/10.7554/eLife.40630.001

## Introduction

**\*For correspondence:**
mfoxhunter@gmail.com

**Competing interests:** The authors declare that no competing interests exist.

Evolution was once thought to work exclusively as a gradual process (*Darwin, 1859*), where speciation followed the accumulation of divergent traits primarily in isolated populations (*Carroll et al., 2007*). Yet decades of research have transformed this model of evolution into one that emphasizes the importance of behavior (*Lapiedra et al., 2018*; *Grant and Grant, 2010*), while accommodating not only rapid phylogenetic and phenotypic diversification (*Rudh et al., 2011*; *Uy and Borgia, 2000*; *Kettlewell, 1956*), but also speciation with gene flow (*Servedio, 2016*; *Servedio, 2004*; *Kopp et al., 2018*). Theory predicts that such instances of rapid speciation may occur provided there is sufficient phenotypic disparity between diverging population to erect reproductive barriers, but this implies that other factors that give rise to phenotypic variation also play an important role in this process. One of these factors is an organism's physiology, which provides the functional architecture that defines how phenotypes can evolve and/or emerge anew. However, studies rarely explore the role of certain physiological systems in the process of sympatric speciation, creating a gap in our understanding of modern evolutionary biology.

A major way that physiology influences evolution is through constraint, due to numerous trade-offs between systems or processes that function in a mutually exclusive manner (*Roff and Fairbairn, 2007*; *Alexander, 1985*). Skeletal muscle performance is a prime example of this phenomenon: its design entails performance traits that are inversely related to each other, such as between speed and endurance (*Vanhooydonck et al., 2014*; *Wilson et al., 2002*). At the same time, skeletal muscle also actuates many sexually-selected behaviors that act as the basis for mate choice (*Goller and Suthers, 1996*; *Garcia et al., 2012*; *Fuxjager et al., 2017a*). This suggests that performance trade-

offs inherent to muscle itself can impede the formation of reproductive barriers, particularly in cases where behavioral divergence should mediate speciation. Evaluating this idea requires a simultaneous examination of a species' evolutionary history, as well as whole-organism and muscle performance. Few studies unite all three perspectives, however, leaving an unclear relationship between animal design, behavior, and evolutionary process.

Here, we address this issue by exploring how performance trade-offs borne out via skeletal muscle shaped the evolution of sexual display behavior during a rapid speciation event. We study a small genus of Neotropical birds called bearded manakins (Aves: Pipridae). This group last shared a common ancestor roughly 300,000 years ago (*Brumfield et al., 2008*; *Tello et al., 2009*), before splitting into four species: white-bearded manakins (*M. manacus*), golden-collared manakins (*Manacus vitellinus*), white-collared manakins (*M. candei*), and orange-collared manakins (*M. aurantiacus*). We specifically examine how disruptive selection operates on one of the bird's main courtship signals—the roll-snap—to generate phenotypic divergence sufficient to facilitate assortative mating (*Servedio, 2016*). Males produce this behavior by rapidly hitting their wings together over their back, generating a mechanical trill-like sound (*Fuxjager et al., 2017a*). The result is a repetitive rhythmic signal, which is a common signal type in animal courtship (*Mowles and Ord, 2012*). These displays convey information through multiple component signals, but one of particular importance in reshaping a receiver's behavior is signal speed (*Schuppe and Fuxjager, 2018*; *Ord et al., 2007*; *Podos, 2001*). As such, we hypothesize that speed is an important feature of roll-snap performance. Although the degree to which roll-snap speed itself is sexually selected remains unknown, previous studies in golden-collared manakins show that females discriminate millisecond-scale differences in other displays and prefer to mate with males that display faster (*Barske et al., 2011*). Therefore, the speed with which a male hits his wings together likely functions as an important courtship signal that is subject to strong sexual selection. If so, then small changes in female preferences for speed may provide the selection pressure necessary for display divergence. However, disruptive selection may not be able to freely drive sister species to exhibit opposite patterns of directional selection for this trait, largely because the speed with which skeletal muscles control wing movements is constrained by important performance trade-offs, such as one between speed and endurance (*Vanhooydonck et al., 2014*; *Wilson et al., 2002*). This leads to the question: what role did intrinsic muscle performance trade-offs play in the rapid diversification of these species?

## Results and discussion

As a first step toward answering the question posed above, we evaluated whether speciation of bearded manakins occurred under sympatry, a special case of speciation with gene flow (*Kopp et al., 2018*). We accomplished this using phylogeographic models (*O'Donovan et al., 2018*; *Lemmon and Lemmon, 2008*; *Pagel et al., 2004*) to reconstruct the ranges of ancestral *Manacus* populations (*Figure 1a*). The model output (*Figure 1b*) shows that the bearded manakins' most recent common ancestor occupied a geographic range similar to that of the western population of extant white-bearded manakins, giving rise to this species before migrating west across the Isthmus of Panama (*Video 1*). Golden-collared manakins were the next to split off, emerging in a geographic range that overlapped between 49% and 73% (95% credible intervals) with that of the final common ancestor to both white-collared and orange-collared manakins. This wide-ranging ancestral overlap is therefore consistent with speciation occurring in sympatry, typically considered to occur when ranges overlap ≥25% (*Seddon, 2005*; *Cooney et al., 2017*; *Tobias et al., 2014*). Furthermore, because modern bearded manakins are mostly allopatric (*Figure 1c*), geographic isolation of these birds likely followed their diversification, rather than preceding it. This is consistent with previous work in golden-collared and white-collared manakins, which hybridize readily in a small region of range overlap in Panama (*Parsons et al., 1993*). In the hybrid zone, directional selection for plumage ornaments is strong enough for golden-collared manakin traits to cross the hybrid zone and introgress into the white-collared manakin population (*Parsons et al., 1993*; *McDonald et al., 2001*; *Brumfield et al., 2001*). Moreover, whole-genome analysis of the two species finds high rates of introgression and genetic differentiation at a wide range of unrelated loci, which is consistent with an evolutionary history characterized by reproductive isolation following adaptive divergence in

**Figure 1.** Evolutionary history of the bearded manakins. (a) From a molecular phylogeny that represents our most up-to-date understanding of the group's evolutionary history (see Materials and methods). (b) Ancestral state reconstruction of range polygons. 95% credible intervals are denoted by transparent outer edges (maximum extent) and the dotted lines (minimum extent). Golden-collared, white-collared, and orange-collared manakins arose from ancestors (2 and 3) that overlapped across 49% to 73% [95 CI] of their range. (c) Present-day species ranges are geographically isolated from one another.

DOI: https://doi.org/10.7554/eLife.40630.002

The following figure supplement is available for figure 1:

**Figure supplement 1.** We reconstructed ancestral ranges by computing the coordinates for 12 index points on the range maps of each manakin species on our phylogeny.

DOI: https://doi.org/10.7554/eLife.40630.003

**Video 1.** Animated reconstruction of ancestral range movements in *Manacus* species. The video was created by first importing the vector images of reconstructed range approximations as vector graphics into Adobe Animate CC. We then used the shape tween function to model how range approximations shifted between their reconstructed node states.

DOI: https://doi.org/10.7554/eLife.40630.004

various traits (*Parchman et al., 2013*). Thus, mechanisms other than those linked to geographic isolation likely help drive rapid speciation in the genus.

If courtship display divergence played a role in speciation, then species arising with gene flow should exhibit different display phenotypes. We therefore examined species differences in the acrobatic roll-snap display by analyzing acoustic recordings of roll-snaps for each species (*Table 1*), measuring both signal speed (snaps $s^{-1}$) and length (total number of snaps). Speed differed significantly among the species (*Figure 2a*; $F_{3,146}=9.203$, p<0.0001), with golden-collared manakins producing the fastest roll-snaps, white-bearded and white-collared manakins (no ancestral overlap) both producing intermediate speed roll-snaps, and orange-collared manakins producing the slowest roll-snaps. The magnitude of mean differences is relatively small overall (<5 Hz), but also reflects larger differences in the upper range of display speed.

**Table 1.** Sample size breakdown from citizen-scientist audio recordings of wild bearded manakins performing the roll-snap display.

All recordings were obtained from Xeno Canto (XC) or the Macaulay Library of Natural Sound (LNS). For accession information and metadata, please see source data file (*Figure 2—source data 1*). For each species, we were able to collect data from more individuals than there are audio recordings. This is because most high-quality recordings captured displays from two or more males displaying at different distances from the recordist, which were clearly distinguished by amplitude.

| Species | # Recordings | # Individuals | # Displays (±1 SEM) per individual |
|---|---|---|---|
| Golden-collared manakin | 21 | 34 | 4.5 ± 0.53 |
| White-collared manakin | 16 | 24 | 3.1 ± 0.51 |
| Orange-collared manakin | 9 | 25 | 2.6 ± 0.46 |
| White-bearded manakin | 40 | 76 | 3.4 ± 0.42 |

DOI: https://doi.org/10.7554/eLife.40630.007

For example, the fastest roll-snap observed in white-collared manakins (61 Hz) falls within the interquartile range of the golden-collared manakin's speed distribution [55.9,61.1]. Because phenotypic constraints typically impact trait evolution by restricting the extremes of a distribution (*Wilkins et al., 2013*), this suggests that shifting constraint space may underlie differences in speed. At the same time, we found that roll snap length in all four species was statistically indistinguishable (*Figure 2b*; $F_{3,153}=0.229$, p=0.876). Therefore, the roll-snap diverged among species exclusively in terms of signal speed (i.e. snap repetition rate), but not in terms of the total number of snaps. Therefore, if females attend to differences in roll-snap speed as they do with other signals (*Barske et al., 2011*), conspecific mate preference should be reinforced by divergent roll-snap speeds (*McDonald, 1989*). Our data are thus consistent with a model in which phenotypic divergence among sympatric populations plays a role in supporting assortative mating (*Servedio, 2016*; *Kopp et al., 2018*).

Evolutionarily modifying a signal to become faster can sometimes incur costs to other performance attributes, such as endurance (*Van Damme et al., 2002*; *Vanhooydonck et al., 2001*; *Reidy et al., 2000*, c.f. *Sorci et al., 1995*). Divergence in the roll-snap may therefore be constrained by these trade-offs, especially in species that evolve faster—but equally as long—roll-snaps. To examine whether this evolutionary impasse is present, we used quantile regression to test for an inverse relationship between the speed and length of each species roll-snap (*Cade and Noon, 2003*; *Koenker and Machado, 1999*; *Miles et al., 2018*; *Wilson et al., 2014*). The mere existence of such a relationship would indicate that a given species was signaling near (or at) its phenotypic limit with respect to these measures of performance. Indeed, at the uppermost testable quantile (τ = 0.9), we detected this trade-off in both golden-collared (*Figure 3a*; t = -4.03, p<0.001) and white-collared manakins (*Figure 3b*; t = -2.76, p=0.007), but not in white-bearded (*Figure 3c*; t = -0.36, p=0.722) and orange-collared manakins (*Figure 3d*; t = -0.70 p=0.490). Roll-snaps are therefore differentially constrained across species.

Even among two species undergoing a performance constraint between speed and endurance, there can be variation in terms of the trade-off's steepness (slope), as well as the proportion of individuals in the population that encounter it. If the trade-offs we uncover are the result of endurance costs, then the negative linear relationship between speed and length should be steeper in the species with the faster display (i.e. golden-collared manakins, see *Figure 2a*). This was indeed the case ($F_{1,230}=22.1$, p=0.001), supporting the notion that performance costs to speed become greater as selection drives the evolution of the signal further into the constrained trait-space. Similarly, because golden-collared manakins have a faster display overall than white-collared manakins (post-hoc: z = 2.5, p=0.038), more individuals should produce displays fast enough to encounter endurance constraints. Our data again support this idea, as nearly all golden-collared manakins (top 80% of the population) experience a cost to display speed (*Table 2*), given the signal's length (*Figure 3a*; at τ = 0.2: t = -2.95, p=0.004), whereas only the top 20% of white-collared manakins display fast enough to experience such a cost (*Figure 3b*; at τ = 0.8: t = -2.33, p=0.028). Thus, for these two species, the rate at which an individual snaps its wings together to display is influenced by the

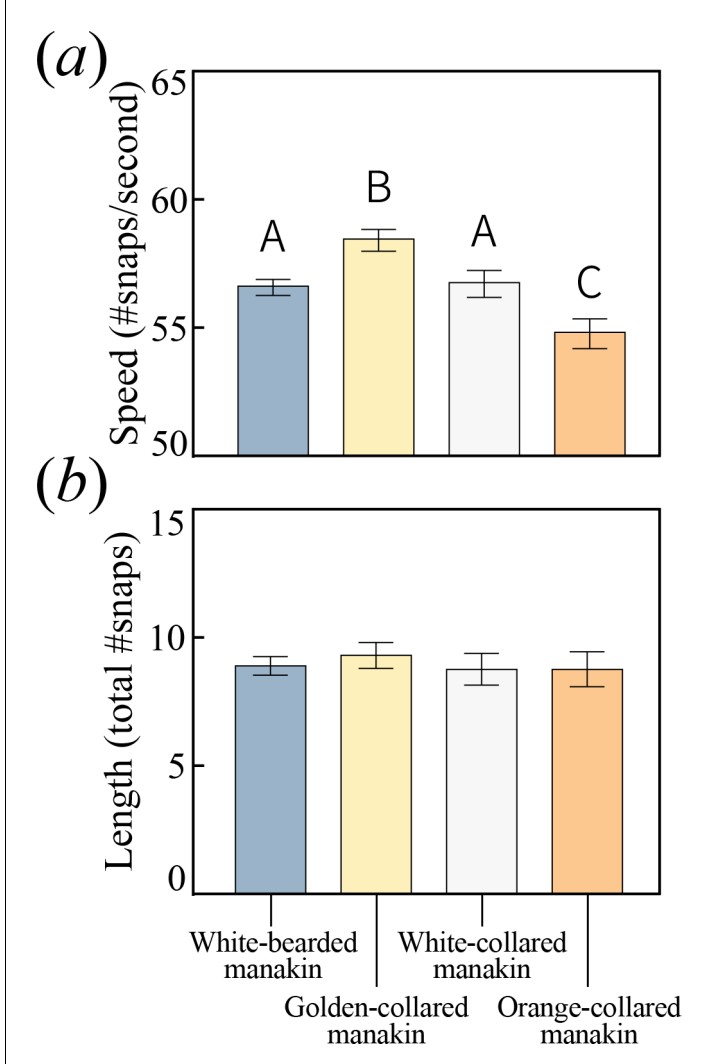

**Figure 2.** All bearded manakins use a unique wing-snapping display, called the roll-snap, for courtship. This signal has phenotypically diverged in terms of (**a**) speed, or the rate (snaps sec$^{-1}$) at which an individual repeatedly hits its wings together above its back (p<0.05, with statistically significant differences between groups denoted by different letters atop the bars). (**b**) There is no apparent divergence in the display's duration in terms of the total number of snap events within a single roll. Values plotted are estimated marginal means ± 1 SEM.

DOI: https://doi.org/10.7554/eLife.40630.005

The following source data is available for figure 2:

**Source data 1.** Data_acoustics.txt: Measurements of roll-snap speed and length obtained from audio recordings of displaying birds.

DOI: https://doi.org/10.7554/eLife.40630.006

inverse relationship between speed and endurance, with the effect of this constraint appearing greater in golden-collared manakins. This may be a consequence of the fact that golden-collared manakins have a faster roll-snap than the other species, but without a change in overall display length. Their roll-snap speed appears to have diverged just enough to maintain the signal's length, with trade-offs consequentially appearing at the level of the individual.

The origin of behavioral trade-offs may lie in the physiological and/or morphological underpinnings of display production (*Podos, 2001*; *Derryberry et al., 2012*). For the roll-snap, skeletal muscle performance may be one of the limiting factors, considering that muscle tissue itself is thought to be the origin of the speed-endurance trade-off (*Vanhooydonck et al., 2014*; *Wilson et al., 2002*;

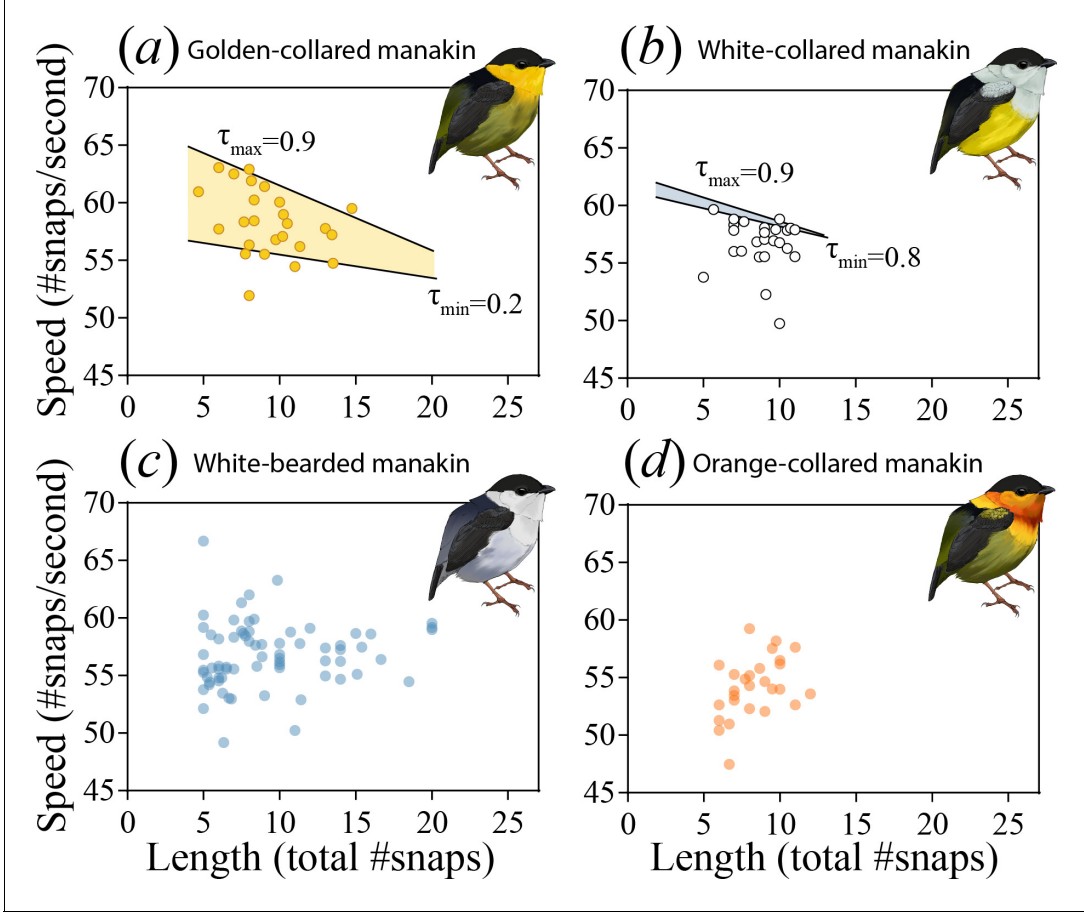

**Figure 3.** The roll-snap display phenotype can be characterized as a bivariate distribution by plotting speed (# of snaps s$^{-1}$) as a function of length (total snaps in a roll) for (a) golden-collared, (b) white-collared, (c) white-bearded, and (d) orange-collared manakins. Values are individual means computed from multiple roll-snap displays. (a) Golden-collared manakins and (b) white-collared manakins both have a significantly negative upper-bound ($\tau$ = 0.9, p<0.05) to the distribution, which is consistent with a performance constraint on speed. (a) For the golden-collared manakin, the negative bound extends continuously into the 20th quantile, thus impacting 80% of the population (at $\tau$ = 0.2: t = -2.95, p=0.004). (b) By contrast, in white-collared manakins, this constraint is only present at the uppermost end of the distribution (80th quantile and above; $\tau$ = 0.8, t = -2.33, p=0.028).
DOI: https://doi.org/10.7554/eLife.40630.008

The following figure supplement is available for figure 3:

**Figure supplement 1.** Roll-snap display length (# snaps) as it predicts speed (#snaps/second) when evaluated at every quantile ($\tau$) between 0.1 and 0.9 at intervals of 0.01.
DOI: https://doi.org/10.7554/eLife.40630.009

*Bottinelli et al., 1994*). In bearded manakins, the *scapulohumeralis caudalis* muscle (SH, *Figure 4a*) is the primary actuator of the roll-snap—it contracts when a male's wings are raised vertically above its back, causing the wrists to collide (snap) over the axial mid-line (*Fuxjager et al., 2017a*). Otherwise, the SH is a flight muscle, acting as a 'strut' during powered locomotion by rotating and retracting the wing (*Dial, 1992*). Past work shows that bearded manakins maintain an extremely fast SH, likely due to selection for a fast display (*Barske et al., 2011*; *Fuxjager et al., 2016a*; *Fuxjager et al., 2017b*); thus, one might expect that the evolution of SH speed contends with a decrease in endurance. To test this idea, we quantified muscle performance first in the golden-collared manakin (the fastest species) and then in the white-collared manakin. Although the white-collared manakin was not the species with the slowest roll-snap, we chose it as a basis of comparison due to the well-studied evolutionary history between these two populations. White-collared and golden-collared manakins are the only pair of *Manacus* species that are well known to hybridize today (*McDonald et al., 2001*), which means that signal divergence is likely important to mediating reproductive barriers in the present day.

**Table 2.** Quantile regression results summary for the roll-snap performance distribution of golden-collared and white-collared manakins, where the display length is evaluated as a predictor of display speed at different quantiles (τ) of the distribution. A significantly negative slope (p<0.05; values adjusted to control the false discovery rate) suggests that there is a trade-off between speed and endurance at a given quantile. Here we present representative models from every 10th quantile (τ interval = 0.1), but we also characterized the performance distribution at a finer-grained scale (see *Figure 3—figure supplement 1*) at every whole quantile (τ interval = 0.01) between 0.1 and 0.9. We did not extend the analysis below the 10th or above the 90th quantile because quantile regression performs poorly at extreme portions of the distribution for smaller datasets (n < 1000).

| Quantile (τ) | Golden-collared manakin | | | | White-collared manakin | | | |
|---|---|---|---|---|---|---|---|---|
| | Slope | s.e. | t-value | p-value | Slope | s.e. | t-value | p-value |
| 0.1 | −0.13 | 0.11 | −1.18 | .239 | −0.02 | 0.58 | −0.03 | .976 |
| 0.2 | −0.20 | 0.07 | −2.95 | .004* | 0.03 | 0.51 | 0.06 | .976 |
| 0.3 | −0.31 | 0.07 | −4.13 | .0001* | −0.04 | 0.31 | −0.14 | .976 |
| 0.4 | −0.44 | 0.08 | −5.27 | <0.0001* | −0.29 | 0.24 | −1.24 | .305 |
| 0.5 | −0.53 | 0.08 | −6.51 | <0.0001* | −0.20 | 0.23 | −0.85 | .512 |
| 0.6 | −0.44 | 0.07 | −6.39 | <0.0001* | −0.19 | 0.15 | −1.27 | .305 |
| 0.7 | −0.58 | 0.03 | −21.69 | <0.0001* | −0.11 | 0.16 | −0.67 | .606 |
| 0.8 | −0.60 | 0.08 | −7.91 | <0.0001* | −0.31 | 0.13 | −2.33 | .028* |
| 0.9 | −0.56 | 0.14 | −4.03 | .0001* | −0.41 | 0.15 | −2.76 | .020* |

DOI: https://doi.org/10.7554/eLife.40630.010

We used in situ recordings of SH twitch speed (*Fuxjager et al., 2016a*; *Fuxjager et al., 2017b*) to test whether skeletal muscle performance underlies differences in display behavior. Accordingly, we measured the percent relaxation of the SH at different stimulation frequencies (30–100 Hz), modeling relative relaxation (0–100%) as a function of muscle stimulation speed (*Figure 4b*). Our results confirmed previous work (*Fuxjager et al., 2016a*) by indicating that golden-collared manakins maintain an extremely fast SH—that is, this muscle relaxes more quickly at high frequencies than it does in white-collared manakins (*Figure 4c*; model difference: $F_{3,49}$=15.2, p<0.001). The muscle therefore also maintains a significantly greater half-relaxation frequency in the golden-collared manakin, which is the maximum frequency at which the muscle contracts and relaxes at 50% of its predicted functional range (*Figure 4d*; $F_{1,49}$=79.8, p<0.001). Next, to determine if the ability of the SH to perform at such speeds is limited by endurance costs, we modeled the tissue's percent relaxation as a function of the stimulation pulse number (1-8) at each given stimulation frequency. This reflects how the muscle's performance changes at different speeds in response to repeated stimulations. We found that the golden-collared manakin SH relaxed between 75% and 100% when stimulated at 50, 60 and 70 Hz (*Figure 5a*). However, when stimulated at ≥80 Hz, its percent relaxation significantly declined with each twitch (*Figure 5a* and *Table 3*; $F_{1,12}$=18.7, p=0.005 at 80 Hz; $F_{1,12}$=33.9, p<0.001 at 90 Hz; $F_{1,12}$=22.8, p=0.003 at 100 Hz). Indeed, we find that the level of relaxation dropped from around 75–100% during the first contraction to around 20–50% as the stimulation train progressed. By contrast, in the white-collared manakin, percent relaxation of the SH never changed across any of the stimulation trains (*Figure 5b* and *Table 3*). Instead, only overall percent relaxation decreased with increasing frequency (see above, *Figure 4c*). These data therefore suggest that the golden-collared manakin SH experiences a form of 'rapid fatigue' when stimulated at especially fast frequencies. This performance attribute contrasts with fatigue as it is normally discussed, where the latter is typically defined as a decrease in speed and force production over periods of time greater than the fractions-of-a-second that we show herein (*Wilson et al., 2002*; *Allen et al., 2008*). Nonetheless, we expect that the ability to resist this 'rapid fatigue' similarly qualifies as a form of muscle endurance per se (the ability to resist fatigue; *Wilson et al., 2002*).

To further examine how this divergence in SH performance relates to display behavior, we modeled the way that the observed 'rapid fatigue' changed in response to stimulation frequency. As such, we first computed the slopes that describe the relationships between percent relaxation and stimulus number within the different stimulation trains (see above; *Figure 5a and b*). We then plotted these values as a function of overall stimulation frequency, finding that the SH of the golden-

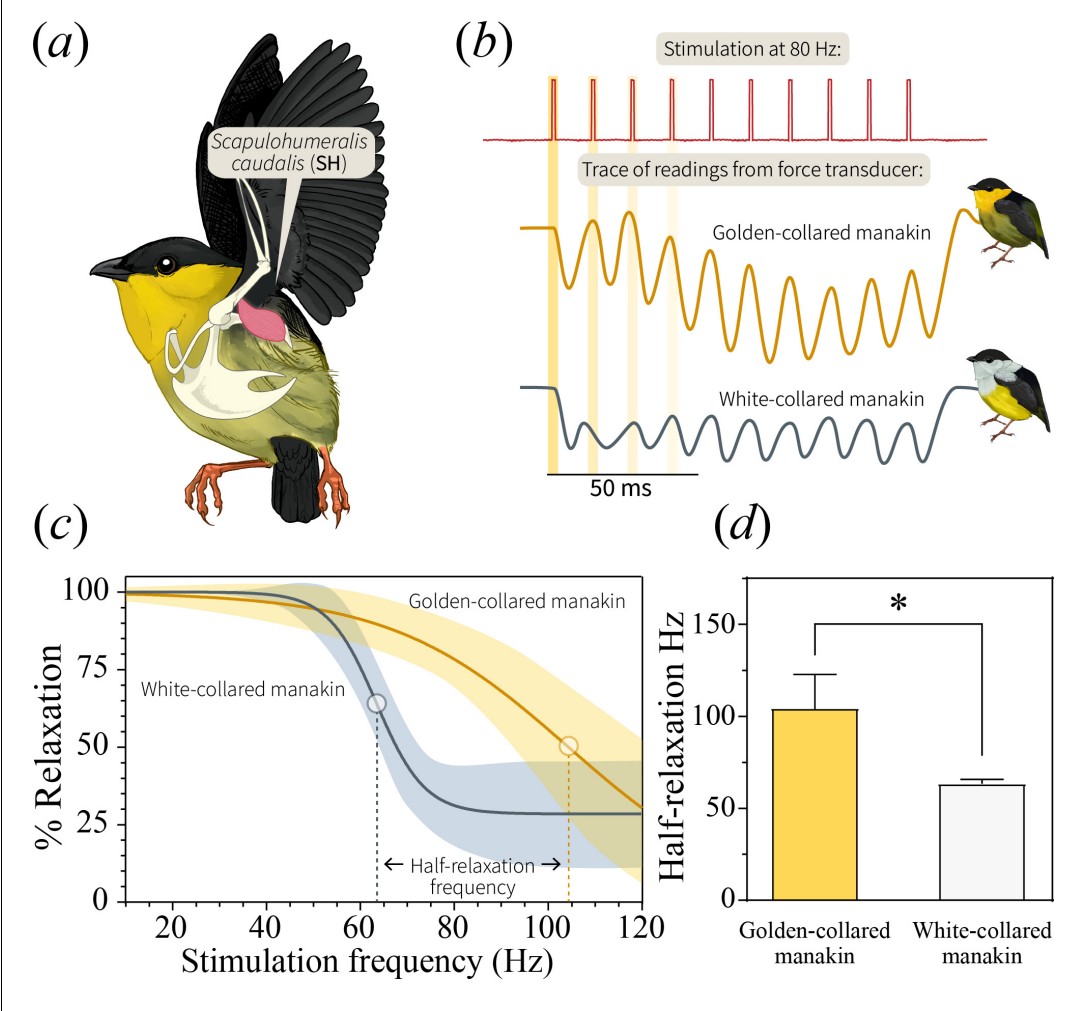

**Figure 4.** In bearded manakins, (**a**) the main humeral retractor, the *scapulohumeralis caudalis* (SH), actuates roll-snap display behavior. Panel (**b**) is a representative in situ recording of twitch speed from the SH when stimulated with an 80 Hz pulse train. Representative recordings from both golden-collared and white-collared manakins are shown. Note that the first four individual stimulation pulses within the entire stimulation train are highlighted yellow, as these data are used for later analyses (See *Figure 5*). (**c**) We subjected the SH to pulse trains of varying frequencies and measured percent relaxation for each contraction in this series. We then averaged these values within each train to generate a plot of percent relaxation as it changes with stimulation frequency. From there, we fit a four-parameter logistic curve to the data that illustrates the twitch dynamics of each species' SH muscle. In these models, the dark solid line reflects the best-fit model ±95% confidence bands (shaded area). (**d**) Our models also allow us to extract an inflection point, which corresponds to the tissue's half-relaxation frequency. This is an index of twitch speed that we can use to compare between species. Bars represent mean ±1 SEM, with the asterisk (*) denoting a significant different between species ($F_{1,49}$=79.8, p<0.001).
DOI: https://doi.org/10.7554/eLife.40630.011
The following source data is available for figure 4:

**Source data 1.** Data_fatigue.txt: Contractile phase measurements collected from each twitch cycle during stimulation of golden-collared and white-collared manakin SH muscles.
DOI: https://doi.org/10.7554/eLife.40630.012

collared manakin experiences a progressively greater decline in performance when stimulated at high speeds (*Figure 5c*; $F_{1,6}$=27.7, $R^2$ = 0.82, p=0.002). This effect is absent in white-collared manakins (*Figure 5c*; $F_{1,7}$=0.15, $R^2$ = 0.02, p=0.710), which instead maintain a constant (fused) state when subjected to the same fast stimulation frequencies. Importantly, for the golden-collared manakin, we show that the upper confidence limit of this regression line crosses 0 at $\approx$60 Hz—a value that represents the relative speed at which the trade-off with endurance should begin to manifest. This roughly correspond to the species' average roll-snap speed, which is 58.4 Hz (*Figures 2a* and *5c*). Thus,

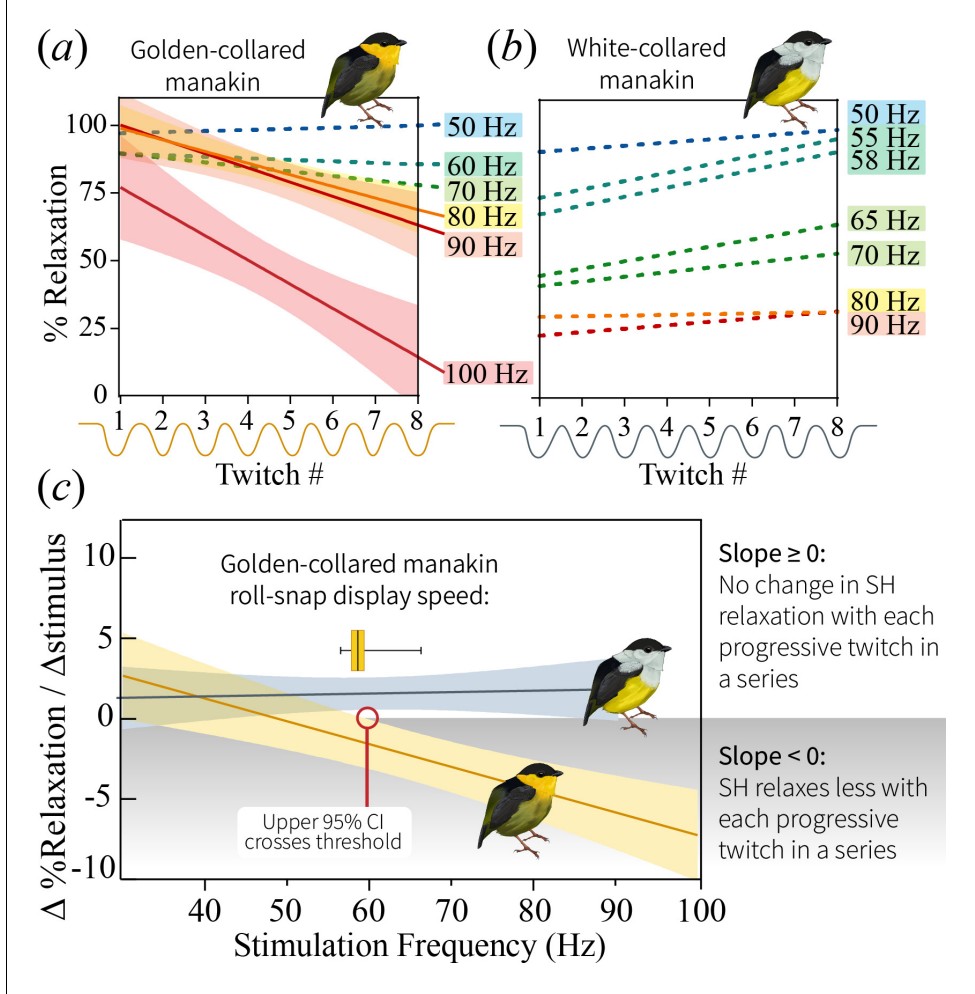

**Figure 5.** Change in percent relaxation of the SH muscle across a single stimulation train at the different stimulation frequencies. (a) SH performance in the golden-collared manakin, whereby SH percent relaxation declines across stimulation trains that are greater than 80 Hz. This is just above the species maximum observed roll-snap speed of 68 Hz. Solid lines represent represent significant regression slopes (p<0.05, β <0), with corresponding shaded areas denoting 95% confidence bands. Non-significant slopes (p>0.05) are indicated by dotted lines. (b) SH performance in the white-collared manakin. Note that all slopes are non-significant (p>0.05), as denoted by the dotted lines; however, the y-intercept appears to progressively decrease as the stimulation frequency increases, suggesting that the muscle fuses at the onset of stimulation and stays that way across the entire stimulation train. (c) Regression lines of the slope of the lines in (a) and (b) plotted as a function of stimulation train frequency. Note that SH performance—as measured by the SH's ability to resist 'rapid fatigue' during high frequency stimulations—declines in golden-collared manakins ($F_{1,6}$=27.7, p=0.004, $R^2$ = 0.82), but not in white-collared manakins ($F_{1,7}$=0.15, p=0.711). Solid lines associated with each species represent the mean change in percent relaxation at a given frequency,±95% confidence bands (shaded areas). The red line indicates the stimulation frequency at which the 95% CI of the SH performance line in golden-collared manakins intersects with the slope = 0 point. In theory, this represents that maximum twitch speed that the muscle can attain without incurring an endurance cost, and it notably corresponds to the species' average roll-snap speed (denoted by horizontal box and whisker plot, where the vertical line is at the mean, shaded box indicates ±1 SEM, and whiskers extend to the species range).

DOI: https://doi.org/10.7554/eLife.40630.013

The following figure supplement is available for figure 5:

**Figure supplement 1.** In the golden-collared manakin, roll-snap displays performed >60 Hz are significantly shorter (t = 6.5, df = 152, p<0.0001) than those ≤60 Hz.

DOI: https://doi.org/10.7554/eLife.40630.014

**Table 3.** Statistical summary for linear models assessing how percent relaxation changes over each successive twitch in stimulation trains of different frequencies.

After evaluating whether each species' SH exhibited a change in performance with repeated stimulations, we also compared the two slope estimates with an F-test. All p-values reported have been adjusted to control for the false discovery rate.

| Stimulation hz | Golden-collared manakin | | | White-collared manakin | | | Species comparison | |
|---|---|---|---|---|---|---|---|---|
| | Slope | F (1,6) | p-value | Slope | F (1,6) | p-value | F (1,12) | p-value |
| 50 | 0.4002 ± 0.3983 | 1.01 | .708 | 1.154 ± 0.6165 | 3.50 | .586 | 1.942 | .189 |
| 55 | | | | 3.098 ± 1.301 | 5.67 | .237 | | |
| 58 | | | | 3.287 ± 1.759 | 3.49 | .586 | | |
| 60 | −1.682 ± 0.8066 | 4.35 | .246 | | | | | |
| 65 | | | | 2.7 ± 1.885 | 2.05 | .767 | | |
| 70 | −0.5629 ± 1.87 | 0.09 | .774 | 1.705 ± 2.215 | 0.59 | 1.00 | 1.44 | .254 |
| 80 | −5.268 ± 1.185 | 19.8 | .017* | 0.2476 ± 1.25 | 0.04 | 1.00 | 18.7 | .002** |
| 90 | −4.298 ± 0.8332 | 26.6 | .013* | 1.268 ± 1.382 | 0.84 | 1.00 | 33.9 | <0.0001*** |
| 100 | −8.949 ± 1.874 | 22.8 | .016* | | | | | |

DOI: https://doi.org/10.7554/eLife.40630.017

sexual selection in golden-collared manakins has likely driven the evolution of the roll-snap to its maximal speed, such that it is produced at the SH's performance limit due to the tissue's intrinsic trade-off with endurance. Further support for this idea comes from a subsequent analysis, in which we found that male golden-collared manakin produce significantly shorter roll-snaps when they are at or above 60 Hz, compared to less than 60 Hz ($\beta = -4.32$, $F_{1,119}=46.7$, $p<0.001$; *Figure 5—figure supplement 1*).

What explains the rapid decline in relaxation with multiple contractions at high speed? Distinct mechanisms may shape the different phases of the contraction-relaxation cycle; thus, we quantified shortening and lengthening periods in twitch recordings that were both (*i*) right above each species' average roll-snap speeds (golden-collared manakin threshold = 60 Hz, white-collared manakin threshold = 58 Hz) and (*ii*) at a high frequency stimulation (80 Hz) (*Figure 6*). Both measures showed considerable variation, but for both species we found no significant changes within the twitch series ($p_{min} = 0.20$; *Figure 6*). This indicates that shortening and lengthening duration remained consistent during the stimulation trains. Nonetheless, in the golden-collared manakin, we know that percent relaxation declines at high stimulation frequencies in this same timeframe. Thus, the overall 'rapid fatigue' response is likely a result of decreased capacity for relaxation, pointing to processes underlying this event as a potential limiter of muscle performance. This may be linked to constraints on calcium buffering and/or re-uptake mechanisms, given they play an important role in setting muscle relaxation rates (*Syme and Josephson, 2002*; *Rome et al., 1996*). Regardless of which cellular mechanism ultimately limits SH performance in the golden-collared manakin, this is consistent with the fundamental framework of selection contending with constraint—the phenotype favored by selection (in this case, display speed) will evolve following directional selection until it is impeded by the underlying rate-limiting mechanism. In this case, it appears that the golden-collared manakin SH has been modified to allow the muscle to sustain rapid contraction-relaxation cycling, but different components of myocytic machinery have likely evolved in a mismatched fashion.

At the cellular level, muscle relaxation requires both rapid cross-bridge detachment and swift calcium cycling (*Syme and Josephson, 2002*; *Rome et al., 1996*). Past work in the golden-collared manakins suggests that the SH can achieve the former, with measures of maximal shortening velocity surpassing the maximum roll-snap speed (*Fuxjager et al., 2016a*). Thus, the factor that limits SH speed for more than two or three twitches—if not related to cross-bridge detachment rates—is likely related to processes governing calcium cycling within the myocyte (*Syme and Josephson, 2002*). This is indicative of evolutionary discordance between different facets of the muscle's contractile machinery, which ultimately put the brakes on sexual selection for a rapid display. Further evolutionary elaboration of the roll-snap speed would likely require modifications to the calcium handling properties, alongside other adaptations for the performance of this signal in bearded manakins

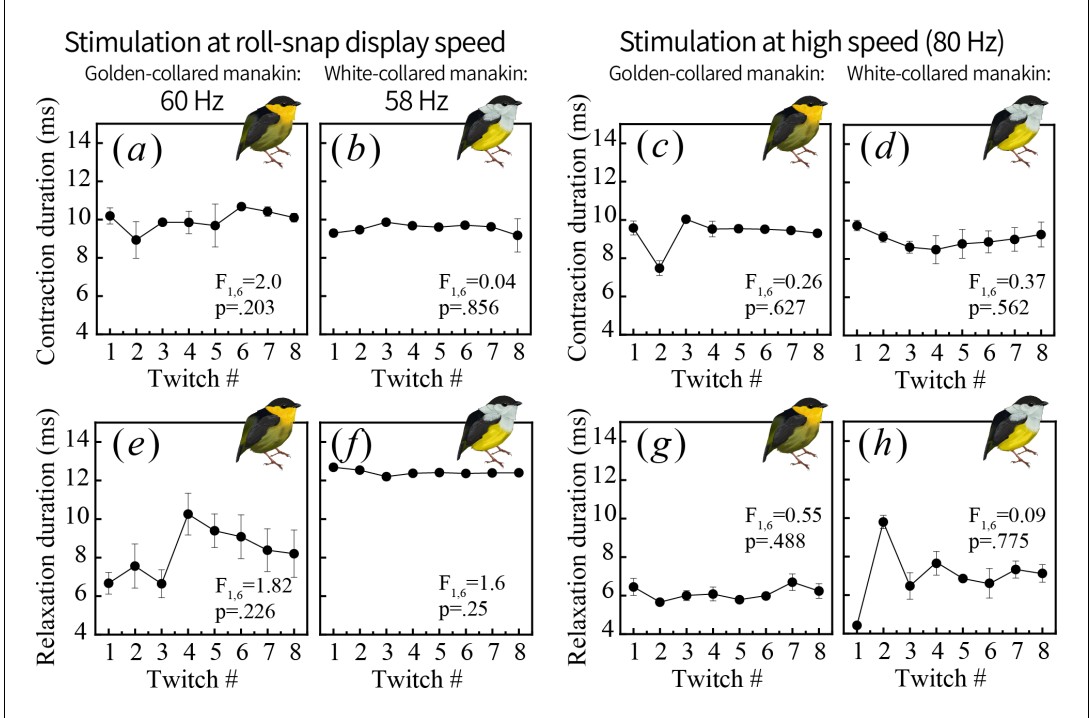

**Figure 6.** One muscle twitch consists of two phases: contraction, or shortening (**a–d**) and relaxation, or lengthening (**e–h**). To better pinpoint the physiological mechanisms generating endurance costs in the golden-collared manakin SH (see *Figure 5*), we measured the duration of each phase and tested whether contractile timing also changes over repeated stimulations administered near the species' roll-snap display speed (**a, b, e, f**) or at a high speed of 80 Hz (**c, d, g, h**). We found no change throughout the stimulation series for any measure, which means the observed decline in percent relaxation shown in *Figure 5* occurs independently of shifts in contractile timing.

DOI: https://doi.org/10.7554/eLife.40630.015

The following source data is available for figure 6:

**Source data 1.** Data_twitchspeed.txt: Twitch speed measurements collected from muscle recordings of golden-collared and white-collared manakin SH muscles.

DOI: https://doi.org/10.7554/eLife.40630.016

(*Friscia et al., 2016*; *Fuxjager et al., 2016b*; *Fuxjager et al., 2016c*). Notably, limitations to muscular calcium handling are not typically associated with the speed-endurance trade-off, which is often considered to result from muscle fiber type composition (*Rivero et al., 1993*; *Komi, 1984*; *Esbjörnsson et al., 1993*). This discrepancy is likely the result of differences between the so-called 'rapid fatigue' we observe and the more standard muscle fatigue that occurs when performance is depleted after prolonged muscle use.

## General discussion

This study provides a blueprint that outlines how trade-offs in skeletal muscle performance influence a rapid sympatric speciation event. We use bearded manakins, which are Neotropical birds that separated into distinct species relatively recently. Our results show that the species in this complex evolved in a geographic scenario characterized by considerable range overlap, which

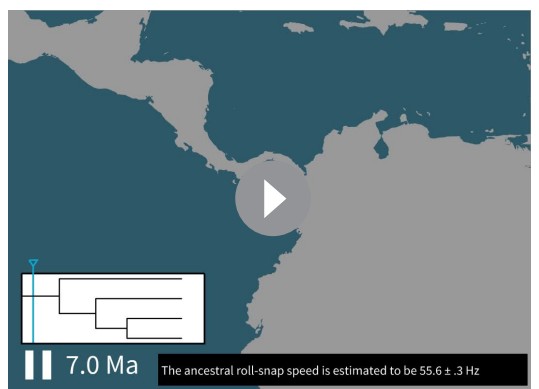

**Video 2.** A slowed-down version of *Video 1*, with additional written information contextualizing how the roll-snap display evolved alongside ongoing range shifts and species divergence.

DOI: https://doi.org/10.7554/eLife.40630.018

means that that rapid speciation was likely facilitated by strong assortative mating (*Servedio, 2016*; *Kopp et al., 2018*). Bearded manakin divergence took place after ancestral populations migrated northwest out of Amazonia, across the Andes, and along the Isthmus of Panama. When the golden-collared manakin first diverged from its shared common ancestor with white- and orange-collared manakins, its range spanned the entire isthmus. As the species split, their roll-snap speeds also diverged for the three taxa that were ancestrally sympatric, and their ranges also began contracting to generate the (mostly) allopatric configuration seen today (*Video 2*). Specifically, we found that the golden-collared manakin's roll-snap increased in speed, presumably at the same time as its SH was modified to support the especially rapid contraction-relaxation cycling that was necessarily to generate the movements that make up this behavior. The underlying driver for signal and range differentiation may have been shifts in female preferences for male courtship display performance (*Kopp et al., 2018*; *Servedio and Burger, 2014*). Alternatively, emergent differences in roll-snap speed may have supported conspecific mate preference independently of directional selection for speed. This may explain why orange-collared manakins, which are not known to hybridize with golden-collared manakins, uniquely exhibit a sharp decrease in roll-snap speed as their range moved south of the sympatric ancestral range. By contrast, the white-collared manakin (which did not undergo a large degree of speed differentiation) still interbreeds with golden-collared manakins where their ranges overlap. It also remains possible that other aspects of the roll-snap phenotype not assessed here—for example, the display's amplitude or modulation thereof—may play a similar role in supporting divergence, but this remains to be seen.

During the parallel differentiation of species, plus their ranges and display phenotypes, the golden-collared manakin alone exhibited an increase in roll-snap speed. Such a change should incur some cost to the signal's length, but we found no matching macroevolutionary shift in roll-snap length. Instead, a trade-off between speed and endurance is apparently borne out at the population level (rather than a macroevolutionary one), where only the species that evolved faster displays experience a trade-off between speed and length. We then show that this evolutionary pattern is explained by contraction-relaxation cycling kinetics in the skeletal muscle that actuates the roll-snap display—namely the SH. Thus, the species with the fastest display, the golden-collared manakin, also has the fastest SH. Further analysis of SH performance, however, reveals that this same bird also undergoes a decline in percent relaxation over repeated contractions at speeds greater than 60 Hz, which corresponds to the species' average roll-snap speed. This helps explain why the SH appears to be faster than the bird's actual display—that is, an intrinsic trade-off between muscle speed and endurance (length) blunts any potential effect of sexual selection for a faster display. Finally, we separately examined whether this endurance 'cost' to speed influences the duration of contraction and relaxation within a stimulation series, and we find that the timing of these two variables remain constant across multiple stimulations. Importantly, these effects occur despite the muscle showing an overall decline in percent relaxation, which points to the process of relaxation in and of itself as the origin of phenotypic constraint on roll-snap evolution.

Altogether, our data show how muscle physiology can influence the process by which divergent selection drives phenotypic evolution to support speciation with gene flow. Indeed, for the nascent radiation of tropical bearded manakins, this processes likely partially hinges on changes to an acrobatic mating dance that inextricably linked to the muscular control of behavioral outflow. Thus, if selection on this display is to proceed, it must contend with constraints imposed by this tissue's design. We provide a glimpse into how this might occur, while also showing that additional changes are necessary to the muscle for selection on the golden-collared roll-snap to proceed further. Of course, other traits like plumage ornamentation have also diverged due to sexual selection (*McDonald et al., 2001*), although recent work suggest sexual dichromatism alone is not likely to facilitate species coexistence or speciation in sympatry (*Cooney et al., 2017*). Nonetheless, our study supports the idea that organism's physiological design is critical for creating the landscape in which phenotypic evolution occurs, and therefore contributes to the process by which species diverge.

## Materials and methods

### Biogeographical models

To explore whether bearded manakins evolved from sympatric ancestral populations, we computed percent overlap between range polygons reconstructed with phylogeographic models in BayesTraits V3 (*Pagel and Meade, 2007*) using a molecular phylogeny from Leite et al., (In preparation). The phylogeny was a maximum likelihood estimate based on a dataset of 2237 ultra-conserved element loci (*Faircloth et al., 2012*) that included almost all manakin species. Branch lengths of the entire phylogeny were made ultrametric using non-parametric rate smoothing (*Sanderson, 1997*) as implemented in TreeEdit v. 1.0 (*Rambaut and Charleston, 2002*).

The biogeographic model accepts two-dimensional geographic coordinates (longitude and latitude) as input variables and then reprojects them to allow for continuous reconstruction of geospatial points under Brownian Motion (BM) (*Lemmon and Lemmon, 2008*; *Lemey et al., 2010*). The model is typically used to reconstruct the location of a single index point (e.g. centroid) for a target ancestor (*Walker and Ribeiro, 2011*), but systematically modeling multiple points allows for detailed inference of ancestral movements and ranges (*O'Donovan et al., 2018*). Here, we used the latter approach to generate range estimates with robust error approximations for the common ancestors of extant *Manacus* species.

Just as supplying geographic centroids (midpoint longitude and latitude) to this model allow for estimation of ancestral range centroids, different index points (such as maximum longitude, maximum latitude) can also be estimated under the same framework. Because we were interested in characterizing the degree to which species arose in sympatry, we first used ArcGIS 10.5 (ESRI) to compute four baseline index points the described the absolute range extent for each species (shapefiles courtesy of BirdLife International [*Ridgeley et al., 2012*]): (i) the point of maximum latitude, (ii) minimum latitude, (iii) maximum longitude, and (iv) minimum longitude. These act as the foundation for reconstructing range extent, as they characterize how far ancestral ranges extended along that NESW axis. However, only using these extrema would reconstruct ancestral ranges without considering how interior range contours are different between species. For this reason, we computed an additional four index points using the ArcToolbox function 'Minimum Bounding Geometry', which computes and displays the smallest quadrangle that contains a given polygon (*Figure 1—figure supplement 1*). On each output rectangle, we superimposed both diagonals and midlines and used the eight radial intersection points of the original shapefile with these midlines to generate eight additional points. In the end, we supplied 12 datasets—one for each geographic point— to our phylogeographic models to reconstruct ancestral ranges.

To obtain estimates with robust error margins, we evaluated the models with Markov chain Monte Carlo (MCMC) (*Pagel and Meade, 2007*), sampling the posterior every 10,000 generations from a 1 million-generation chain after a 100,000 generation burn-in. The BayesTraitsV3 autotuner was sufficient to generate well-mixed chains with acceptance rates of 30–36%. Plotting the posterior distribution for each set of points in ArcGIS allowed us to visualize the minimum and maximum bound estimates for ancestral ranges at each point. We conservatively computed our credible intervals for the posterior as the 5th and 95th percentile of the distribution. Finally, we determined percent overlap (±95% CI) for common ancestors as the overlap area divided by the area of the smaller ancestral range. This is a standard approach used to characterize two populations as sympatric and allopatric, where a percent overlap >20–25% is usually considered to be sympatric (*Seddon, 2005*; *Cooney et al., 2017*; *Tobias et al., 2014*; *Pigot and Tobias, 2013*).

### Acoustic data collection

We downloaded audio recordings (n = 119) of adult male golden-collared manakins (*Manacus vitellinus*), white-collared manakins (*M. candei*), orange-collared manakins (*M. aurantiacus*) and white-bearded manakins (*M. manacus*) displaying at leks in Central and South America (*Table 1*). Each species occupies a relatively small geographic range in Central America, with the exception of white-bearded manakins. This taxon comprises geographically separated subspecies that split off from the ancestral *M. manacus* to occupy isolated ranges along South America's Pacific coast (*M. m. purus*), Atlantic coast (*M. m. gutturosus*), and interior Amazonia (*M. m. interior* and *purus*) (*Brumfield et al., 2008*). Because these populations evolved separately from the bearded manakins in Central

America, we did not include any recordings from these populations or geographic regions in our analysis. All recordings were collected by researchers and citizen-scientists (*Table 1*) and archived at Xeno Canto (http://xeno-canto.org) and the Macaulay Library of Natural Sounds (Cornell University).

We first examined recording metadata, if present, to determine the number of individuals displaying in the recording (see *Table 1* for sample breakdown; all sample sizes were determined by the number of available recordings to analyze). In the absence of this information or other clear indicators of individual identity (e.g. if one bird is nearby and the other distant, or if one is picked up heavily in the right channel and the other by the left), we considered each recording to represent one individual to avoid pseudo-replication. The roll-snap is a broadband acoustic display, which is best characterized only in terms of speed (#snaps/s) and length (number of snaps) (*Miles et al., 2018*). Unlike other acoustic parameters, these measurements should not be influenced by unknown distance between recordist and subject (*Price et al., 2006*; *Miles and Fuxjager, 2018*). We extracted data from all recordings in Adobe Audition CC, where we measured roll-snap duration as the time elapsed between the first snap and last snap in the display. Roll-snap speed is s the number of snaps performed over time; thus, we counted the number of snaps in each display and divided it by display duration to calculate speed.

## Acoustic analyses

We first tested for species differences in roll-snap speed and length used linear mixed models (LMM) using the R package 'nlme,' followed by post-hoc testing in 'glht'. The mixed model approach allowed us to test for species differences in roll-snap speed and length without eliminating multiple observations from individuals, as we included individual identity as a random factor. p-Values from the post-hoc tests were all corrected for multiple testing by controlling for the false discovery rate (*Benjamini and Hochberg, 1995*).

Next, we examined whether roll-snap length predicts speed across different portions of the bivariate speed-length distribution. To do this, we computed the average roll-snap length and speed for each individual and ran a series of quantile regression models with these data in the R package 'quantreg', weighting individuals by the number of displays measured. Quantile regression is functionally analogous to ordinary least squares (OLS) regression, but OLS regresses through the mean, whereas quantile regression can operate on any specified quantile ($\tau$; a value between 0 and 1 that specifies which portion of the distribution the model will examine). As such, running the model at $\tau = 0.5$ is regression through the median, while increasing values of $\tau$ represent regressions at increasingly higher portions of the roll-snap speed distribution. In ecology and evolution, quantile regression is used to characterize trade-offs between complex traits when multiple factors may influence variation in a response that is ultimately limited by the predictor at the uppermost portion of the distribution (*Cade and Noon, 2003*; *Miles et al., 2018*). When this occurs, the response-predictor plot no longer resembles a linear function that can be characterized by OLS regression; instead, the distribution is triangular, with lower values of x achieving a wide range of y values, and vice-versa. If a trade-off exists, the upper bound ($\tau = 0.9$ or higher, depending on sample size [*Wilson et al., 2014*]) to this triangle should have a significantly negative slope. In the case of roll-snap displays, a negative upper bound indicates that roll-snap length negatively predicts speed only with regard to high-speed displays. Because we were interested in comparing the degree to which different populations undergo trade-offs, for each species we ran a series of quantile regressions across the entire performance distribution for each species, starting at $\tau = 0.1$ and ending at $\tau = 0.9$ in increments of 0.01 (*Figure 3—figure supplement 1*). If multiple species exhibited significantly negative upper-bounds to the performance distribution, we then compared the degree of this trade-off using an F-test for slope equivalence at $\tau = 0.9$.

## In situ muscle recordings

Using mist nets, we captured adult male golden-collared manakins (n = 3–5) and white-collared manakins (n = 3) on their breeding territories. Sample sizes for this experiment were determined based on our previous work (*Fuxjager et al., 2016a*) showing that three individuals should be sufficient to reveal biological differences. Work with golden-collared manakins was carried out at the Smithsonian Tropical Research Institute (STRI) in Gamboa, Panama, whereas white-collared manakin work was conducted at La Selva Biological Station in Costa Rica. All birds were captured during the height of

the breeding season (March), when males were actively displaying on their courtship arenas. The muscle twitch speed data for the golden-collared manakin were presented elsewhere for a different species comparison analysis (*Fuxjager et al., 2016a*). However, for this study, we re-analyzed the data in a novel way to assess speed-endurance trade-offs relative to white-collared manakins. This work was approved by all federal, local, and institutional authorities, which includes approval from the Animal Care and Use Committees at STRI, La Selva, and Wake Forest University.

We measured contraction-relaxation cycling speeds in the *scapulohumeralis caudalis* (SH) wing muscle of both golden-collared and white-collared manakins, following methodology described in detail elsewhere (*Fuxjager et al., 2016a*; *Fuxjager et al., 2017b*). These measurements were collected in situ, making it possible to collect twitch speed data from an individual without euthanizing it. Briefly, birds were restrained on a surgical pad and anesthetized with isoflurane (2–4% in $O_2$). We then exposed the SH with a 1 cm incision along the back directly over the muscle. We implanted the tissue with the stripped ends (2 mm) of silver wire electrodes that were connected to the stimulator (Model 2100, A-M Systems, WA). We also attached the muscle to a stainless steel micro-hook (0.1 mm diameter) that was connected to a force transducer (Model FT03, Grass Technology, RI) by a monofilament line. The force transducer was anchored to a heavy stand, which we moved to adjust slack in the line; this allowed us to keep tension on the line between the muscle and force transducer, such that the sensitivity of muscle twitch recordings was preserved without overloading the device. Once the preparation was complete, we placed a drop of normal avian saline (0.9%) over the exposed muscles to prevent tissues desiccation during the recording session. Upon completion of the experiment, we removed the electrodes and micro-hook, closed the incision with Vetbond tissue adhesive, and released the bird back into the wild. Surgical preparations occurred at room temperature, which mirrored the outside ambient temperature ($\approx 30°C$).

The force transducer was connected to an AC/DC strain gauge amplifier (model P122; Grass Technologies) to amplify (5K to 10K) and low-pass filter (3 kHz) the signal. Using the DC input selection, the output signal was recorded on a laptop via an A-D converter (model NI USB-6212; National Instruments, Austin, TX) with AviSoft-RECORDER (v.4.2.22). We administered a series of 10 0.5–0.8 mA pulses to the SH at a range of frequencies (golden-collared manakin: 10 Hz, 20 Hz, 30 Hz, 40 Hz, 50 Hz, 60 Hz, 70 Hz, 80 Hz, 90 Hz, and 100 Hz; white-collared manakin: 30 Hz, 40 Hz, 50 Hz, 55 Hz, 58 Hz, 65 Hz, 70 Hz, 80 Hz, and 90 Hz). After an entire series was finished, we delivered another stimulation train at either 20 Hz (for the golden-collared manakin) or 30 Hz (for the white-collared manakin) to verify that muscle performance had not declined overall during data collection. Indeed, SH percent relaxation did not change between the first and second set of 20 Hz stimulation in the golden-collared manakin (paired $t = 1.0$, p=0.42) or 30 Hz stimulations in the white-collared manakin (paired $t = 1.0$, p=0.42).

## Muscle recording analysis

We extracted data from all muscle recordings following previously published methodology (*Fuxjager et al., 2016a*) using Praat (v.5.4) software. We assessed contraction-relaxation cycling speeds within a stimulus period by measuring the amount relaxation relative to the muscle's unstimulated length. SH relaxation (i.e., from 0% to 100%) was therefore calculated by dividing the measured degree of relaxation by that which was otherwise necessary for full recovery.

First, to assess average levels of SH performance between golden-collared and white-collared manakins, we averaged percent relaxation across the first eight stimulations in a pulse train. This corresponds roughly to the number of wing-snaps that comprise a roll-snap in all species of bearded manakin (see *Figure 2b*). At each stimulation frequency, we administered and measured three stimulation trains and averaged across these technical replicates to generate mean percent relaxation for each individual. We then fit a four-parameter logistic regression model to the data collected from each species. The shape of the curve produced by these models resembles a reverse sigmoid, where the central inflection point is the half-relaxation speed. This is the estimated stimulation frequency at which the muscle contracts and relaxes at 50% within its estimated functional range (*Fuxjager et al., 2016a*; *Allen et al., 2008*). Thus, greater half-relaxation frequencies mean that a muscle can relax at 50% when subject to a higher stimulation frequency, and therefore this measure serves as a common and useful index of overall twitch speed (*Fuxjager et al., 2016a*; *Allen et al., 2008*). To evaluate SH performance between species, we statistically compared the two four-parameter logistic regression models, as well as estimates of half-relaxation frequencies using t tests.

Second, we assessed the degree to which percent recovery changed in the SH across a given stimulation train. Thus, for both species, we computed the best-fit slope from an OLS regression, in which percent relaxation was plotted as a function of stimulation number. We repeated this for each of the different stimulation frequencies (see above). If percent relaxation progressively decreases as a stimulation train progresses (i.e. with repeated stimulations in each train), then the slope of this regression line should be negative. We refer to this phenomenon, where muscle recovery decreases within a stimulation train, as 'rapid fatigue.' By contrast, if percent relaxation does not change as a stimulation train progresses, then the slope of the regression line should be zero. Notably, fusion of the SH at the onset of the stimulation train is denoted by a low y-intercept value (i.e. percent relaxation).

Finally, we tested how the development of 'rapid fatigue' changes as a function of stimulation train speed. This allowed us to explore how such fatigue can change across different levels of muscle performance. We therefore fit another linear model (one for each species) to test whether stimulation frequency predicts the 'rapid fatigue' slopes described above. If the slope of the best-fit line for either of these models is negative, then it indicates that 'rapid fatigue' develops when the muscle is subject to faster stimulation frequencies. If the slope of this best-fit line is zero, then it suggests that no such 'rapid fatigue' develops. However, it is important to note that a slope of zero will occur even when the muscle is fused, because percent relaxation—even if it's a minimal amount—is constant across the different stimulation train speeds.

## Acknowledgements

We thank the staff and administration of STRI and La Selva Biological Station for assistance in facilitating fieldwork, as well as the Autoridad Nacional del Ambiente in Panama and the Ministerio de Ambiente, Energia y Telecomunicaciones in Costa Rica for permit authorization. We thank Mike Ryan and Barney Schlinger for logistical support for field work

## Additional information

### Funding

| Funder | Grant reference number | Author |
|---|---|---|
| National Science Foundation | IOS-1655730 | Matthew J Fuxjager |
| Wake Forest University | N/A | Matthew J Fuxjager |

The funders had no role in study design, data collection and interpretation, or the decision to submit the work for publication.

### Author contributions

Meredith C Miles, Formal analysis, Investigation, Methodology, Writing—original draft, Writing—review and editing; Franz Goller, Formal analysis, Investigation, Methodology, Writing—review and editing; Matthew J Fuxjager, Conceptualization, Resources, Formal analysis, Supervision, Funding acquisition, Investigation, Methodology, Writing—original draft, Project administration, Writing—review and editing

### Author ORCIDs

Meredith C Miles http://orcid.org/0000-0002-7307-0195
Franz Goller http://orcid.org/0000-0001-5333-1987
Matthew J Fuxjager http://orcid.org/0000-0003-0591-6854

### Ethics

Animal experimentation: This study was performed according to the protocol approved by the Institutional Animal Care and Use Committee (IACUC) of Wake Forest University (A16-188).

Decision letter and Author response

Decision letter https://doi.org/10.7554/eLife.40630.022

Author response https://doi.org/10.7554/eLife.40630.023

## Additional files

### Supplementary files

• Transparent reporting form

DOI: https://doi.org/10.7554/eLife.40630.019

### Data availability

All data generated or analyzed during this study are included in the manuscript and supporting files.

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
