## [Decision Letter]

Thank you for submitting your article "Physiological constraint on acrobatic courtship behavior underlies a rapid sympatric speciation event" for consideration by *eLife*. Your article has been reviewed by Eve Marder as the Senior Editor, a Reviewing Editor and three reviewers. The following individual involved in review of your submission has agreed to reveal his identity: Ronald L Calabrese (Reviewer #1).

The reviewers have discussed the reviews with one another and the Reviewing Editor has drafted this decision to help you prepare a revised submission.

Summary:

In this advance, the authors investigate how muscle physiology sets limits on courtship displays during sympatric speciation in bearded manakin birds. This work builds on previous work on the *scapulohumeralis caudalis* muscle of golden-collared manakins that showed that the rapid contraction kinetics of this muscle had evolved enabling courtship displays involving rapid wing movements, while in closely related and distantly related species not employing rapid wing movements in courtship it had not. In this work they show that the golden-collared manakin, which has a higher frequency but equal duration courtship display compared with the sympatrically evolved white-collared manakin has different relaxation dynamics to its muscle that indicate that it has been pushed to a frequency limit by evolution. The experiments with the muscles are carefully done and the results clean. The genetic and geospatial arguments for sympatric speciation are clearly presented. This is a very interesting advance that greatly expands the evolutionary context of the previous study and is a significant advance in its own right. The writing is generally clear and necessary data is presented. It should be of wide interest.

Essential revisions:

1) The author assume that the roll-snap is sexually selected. There is no evidence, in the paper or elsewhere, that the speed of the roll-snaps is sexually selected, however. This is a plausible assumption but not warranted as an assertion because no specific tests were made, i.e. choice tests or correlational studies between roll-snap speed and mating success (as done for the jump-snap display in Barske et al., 2011). There is a difference between species in the muscle properties associated with the signal features but no clear connection with the forces driving speciation. The authors should develop their introduction and discussion accordingly.

2) It remains to be explained why two species (white-collared and orange-collared manakins) reduced the roll-snap speed in the evolutionary scenario presented. Other aspects of the sound, not investigated here, could be important, for example the loudness as it can be related to the force of the clapping. In a similar vein, why were the white-bearded manakin and the orange-collared manakin excluded from the analysis of muscle dynamics? While Figure 3 shows that for these species there is no indication of a performance constraint on speed, still one would expect these birds having lower (particularly the very low orange-collared manakin) display frequency and thus significantly different muscle dynamics. Why was this not explored?

3) It will help readers to bring everything (speciation, trade-off, constraint) together again in the final conclusion. A chronological account (what happened when and why) might do that trick. The video provide might be a perfect supplementary venue for this account with the proper descriptive (audio) accompaniment.

4) The title should mention the organisms used '… in manakins' or at least '…in birds'.

[Editors' note: further revisions were requested prior to acceptance, as described below.]

Thank you for resubmitting your work entitled "Physiological constraint on acrobatic courtship behavior underlies rapid sympatric speciation in bearded manakins" for further consideration at *eLife*. Your revised article has been favorably evaluated by Eve Marder (Senior Editor), a Reviewing Editor, and two reviewers.

The manuscript has been improved but there are some remaining issues that need to be addressed before acceptance, as outlined below:

The authors should address the potential minor error in the video mentioned in the comments of Reviewer 2. Checking/correcting this potential error should be very quick and the manuscript will not require re-review. Otherwise the revision is acceptable.

Reviewer #2:

I think the video contains an error – it states that white-collared manakins snap at 55.4 Hz, while according to Figure 2A, this should be ~56.8 Hz. The value of 55.4 Hz is also at odds with the statement that snapping 'speeds up slightly' from 55.9 in the ancestor of white-collared and orange-collared manakins.

---

## [Author Response]

Essential revisions:1) The author assume that the roll-snap is sexually selected. There is no evidence, in the paper or elsewhere, that the speed of the roll-snaps is sexually selected, however. This is a plausible assumption but not warranted as an assertion because no specific tests were made, i.e. choice tests or correlational studies between roll-snap speed and mating success (as done for the jump-snap display in Barske et al., 2011). There is a difference between species in the muscle properties associated with the signal features but no clear connection with the forces driving speciation. The authors should develop their introduction and discussion accordingly.

We have reframed our argument about sexual selection to make it clear that female preferences for display speed have only been tested generally, but not with respect to roll-snap speed per se.

For example:

Introduction: Rewritten to remove the implication that roll-snap speed is known to be sexually selected. Instead, we make the case for plausible sexual selection while differentiating between what is known (that females discriminate millisecond-scale differences in display performance and mate with males performing other displays at high rates) and unknown (if females directly use roll-snap speed in mate choice). We also cite other studies pointing to speed as an important signal in mate choice.

Results section: When interpreting the divergence in roll-snap speed among sympatric populations, we have again removed any language suggesting we know the signal is sexually selected.

Discussion section: We now provide an alternative interpretation that does not hinge on direct selection for roll-snap speed, but simply emphasizes how signal divergence can support conspecific mate preferences.

Discussion section: In the Conclusions we again make it explicit that our discussion with respect to sexual selection remains hypothetical: “… blunts any potential effect of sexual selection for a faster display.”

2) It remains to be explained why two species (white-collared and orange-collared manakins) reduced the roll-snap speed in the evolutionary scenario presented. Other aspects of the sound, not investigated here, could be important, for example the loudness as it can be related to the force of the clapping.

With a de-emphasis on directional selection for speed in the golden-collared manakin, we now instead highlight the importance of signal divergence more generally in supporting differentiation of sympatric populations. In the Discussion section, this leads to an explicit comparison between white- and orange-collared manakins with respect to the fact that one species (orange-collared) exhibited a steep drop in roll-snap speed while the other did not.

The idea that shifts in amplitude may be important is also an interesting one, which we now consider in the Discussion section. We were unable to test for this here, as we were measuring acoustic recordings taken from multiple equipment setups and unknown distances from the roll-snapping bird.

In a similar vein, why were the white-bearded manakin and the orange-collared manakin excluded from the analysis of muscle dynamics? While Figure 3 shows that for these species there is no indication of a performance constraint on speed, still one would expect these birds having lower (particularly the very low orange-collared manakin) display frequency and thus significantly different muscle dynamics. Why was this not explored?

When designing the study, we decided to focus on the white-collared manakin as a comparison species due to the extensive literature on the white-collared/golden-collared hybrid zone and phenotypic differences between the two species. Because they still hybridize, we hypothesized that these two species would be most likely to diverge in roll-snap display speed (and thus muscle physiology) over the others.

We now address this in the Results section.

3) It will help readers to bring everything (speciation, trade-off, constraint) together again in the final conclusion. A chronological account (what happened when and why) might do that trick. The video provide might be a perfect supplementary venue for this account with the proper descriptive (audio) accompaniment.

Following this suggestion, we added a walkthrough in the conclusion that provides a chronological description of events (Discussion section).

The new video also includes a slower breakdown of evolutionary events with text annotations.

4) The title should mention the organisms used '… in manakins' or at least '…in birds'.

This has been fixed.

[Editors' note: further revisions were requested prior to acceptance, as described below.]

The manuscript has been improved but there are some remaining issues that need to be addressed before acceptance, as outlined below:[…]Reviewer #2:I think the video contains an error – it states that white-collared manakins snap at 55.4 Hz, while according to Figure 2A, this should be ~56.8 Hz. The value of 55.4 Hz is also at odds with the statement that snapping 'speeds up slightly' from 55.9 in the ancestor of white-collared and orange-collared manakins.

This was a small typographical error, which we have now fixed. We also went through and made sure that all of the other values reported in the video are correct, and indeed they are.